# Loss of MER Tyrosine Kinase Attenuates Adipocyte Hypertrophy and Leads to Enhanced Thermogenesis in Mice Exposed to High-Fat Diet

**DOI:** 10.3390/cells13221902

**Published:** 2024-11-18

**Authors:** Krisztina Köröskényi, László Sós, Melinda Rostás, Albert Bálint Papp, Endre Kókai, Éva Garabuczi, Dávid Deák, Lívia Beke, Gábor Méhes, Zsuzsa Szondy

**Affiliations:** 1Division of Dental Biochemistry, Department of Basic Medical Sciences, Faculty of Dentistry, University of Debrecen, 4032 Debrecen, Hungary; kkriszti@med.unideb.hu (K.K.); ekokai@med.unideb.hu (E.K.); 2Department of Biochemistry and Molecular Biology, Faculty of Medicine, University of Debrecen, 4032 Debrecen, Hungary; 3Doctoral School of Dental Sciences, Faculty of Dentistry, University of Debrecen, 4032 Debrecen, Hungary; sos.laszlo@med.unideb.hu (L.S.); rostas.melinda@med.unideb.hu (M.R.); papp.albert@dental.unideb.hu (A.B.P.); 4Department of Medical Chemistry, Faculty of Medicine, University of Debrecen, 4032 Debrecen, Hungary; 5Department of Integrative Health Sciences, Institute of Health Sciences, Faculty of Health Sciences, University of Debrecen, 4032 Debrecen, Hungary; gevi@sph.unideb.hu; 6Laboratory Animal Facility, Life Science Building, University of Debrecen, 4032 Debrecen, Hungary; deak.david@med.unideb.hu; 7Department of Pathology, Faculty of Medicine, University of Debrecen, 4032 Debrecen, Hungary; beke.livia@med.unideb.hu (L.B.); gabor.mehes@med.unideb.hu (G.M.)

**Keywords:** Mer tyrosine kinase, obesity, adipocyte, thermogenesis

## Abstract

Obesity is characterized by low-grade inflammation that originates predominantly from the expanding visceral adipose tissue, in which adipocytes respond to lipid overload with hypertrophy, and consequently die by apoptosis. Recruited adipose tissue macrophages (ATMs) take up the excess lipids and remove the dead cells; however, long-term exposure to high concentrations of lipids alters their phenotype to M1-like ATMs that produce pro-inflammatory cytokines and resistin leading to insulin resistance and other obesity-related pathologies. Mer tyrosine kinase is expressed by macrophages and by being an efferocytosis receptor, and by suppressing inflammation, we hypothesized that it might play a protective role against obesity. To our surprise, however, the loss of Mer protected mice against high-fat diet (HFD)-induced obesity. We report in this paper that Mer is also expressed by adipocytes of both white and brown adipose tissues, and while its activity facilitates adipocyte lipid storage both in vitro and in vivo in mice exposed to HFD, it simultaneously attenuates thermogenesis in the brown adipose tissue contributing to its ‘whitening’. Our data indicate that Mer is one of the adipocyte tyrosine kinase receptors, the activity of which contributes to the metabolic decision about the fate of excess lipids favoring their storage within the body.

## 1. Introduction

Obesity is considered a worldwide epidemic and has been linked to the development of a variety of diseases, such as cardiovascular diseases, type-2 diabetes mellitus, chronic lung and oral infections, and several types of cancer. Multiple lines of evidence demonstrate that it is characterized by low-grade inflammation that originates from the expanding adipose tissue [1]. Inflammation develops predominantly in the visceral fat tissue, which responds to lipid overload by adipocyte hypertrophy and is triggered and maintained by the continuous apoptosis of lipid-overloaded adipocytes [2]. Adipocyte enlargement results from an increase in lipid droplet size, and the adipocyte responds to it by an adaptive cell growth regulated by insulin or insulin-like growth factor 1 (IGF-1) via activating the mechanistic target of rapamycin complex 1 (mTORC1) signaling pathway [3]. Hypertrophic adipocytes release chemoattractants, such as monocyte chemoattractant protein-1 (MCP-1) [4], to recruit bone marrow-derived macrophages (MΦs) for their proper clearance. Initially, the recruited MΦs play beneficial roles in maintaining adipose tissue homeostasis [5,6], not only via efferocytosis but also through buffering lipids by taking up the excess lipids released in the form of extracellular vesicles from both living [7] and dying adipocytes [8]. While adipose tissue MΦs in lean mice have an anti-inflammatory (M2) phenotype, MΦs exposed in the long term to high concentrations of lipids switch their phenotype to the so-called M1-like metabolically activated one characterized by a high rate of lysosomal biogenesis and the production of pro-inflammatory cytokines and resistin [9,10].

Pro-inflammatory cytokines are released by lipid-loaded adipocytes themselves, and their amount is positively correlated with the degree of hypertrophy [11]. Lipid accumulation also alters adipocyte adipokine synthesis, leading to enhanced release of leptin and resistin and to decreased production of adiponectin [12]. Since adiponectin improves insulin sensitivity, while resistin and pro-inflammatory cytokines lower it [13], altogether, these alterations in the adipokine and cytokine levels lead to decreased whole-body insulin sensitivity, to compensatory increases in circulating plasma insulin levels, and later to the development of type-2 diabetes mellitus [14,15,16].

In addition, MΦ-derived tumor necrosis factor (TNF)-α [17,18] and adipocyte-derived leptin [19] also contribute to the cell death induction of hypertrophic adipocytes. As a result, during long-term obesity, more and more hypertrophic adipocytes die, leading to a marked loss of the visceral adipose tissue size, and relocation of its lipid depots to other tissues, such as the liver, where the lipid accumulation results in hepatic steatosis [20,21].

The TAM receptors (Tyro3/Axl/Mer) form one of the families of tyrosine kinase receptors [22,23]. The Mer receptor tyrosine kinase (Mer), a member of this family, is expressed among many others by a broad range of immune cells [24], including tissue-resident MΦs, and its expression is highly induced in bone marrow-derived MΦs during M1 to M2 conversion [25,26]. In MΦs, it contributes to the recognition of phosphatidylserine (PS) on the surface of apoptotic cells in a Gas6- and protein S-dependent manner, and thereby promotes the phagocytic clearance of dead cells and the resolution of inflammation, and facilitates tissue repair after acute injury [26,27,28,29,30,31]. Since the lipid-containing extracellular vesicles released by adipocytes are also PS-positive [8], Mer might also contribute to the lipid vesicle uptake by adipose tissue MΦs [32]. In this study, we tested the hypothesis, by studying Mer-null mice [27] exposed to a high-fat diet (HFD), that by promoting efferocytosis, by attenuating pro-inflammatory cytokine production [27,28], and by taking up lipid-containing extracellular vesicles, MΦ Mer might be positively involved in the regulation of adipose tissue lipid homeostasis. Our findings reported in this paper indicate that the involvement of Mer in adipose tissue homeostasis is more complex than we predicted, because, unexpectedly, Mer is also expressed by adipocytes, and the loss of adipocyte Mer dominates the resulting phenotype.

## 2. Materials and Methods

### 2.1. Reagents

All reagents were obtained from Merck (Darmstadt, Germany), except when indicated otherwise.

### 2.2. Animals and Diets

Eight-week-old male Mer-deficient mice [27] and their wild-type littermates were housed in separate cages (with water and food ad libitum) during the 18 weeks of the feeding experiment. Both wild-type and Mer-null mice were divided into two groups: the HFD group received a high-sucrose/high-fat diet (HFD; 45% kcal% fat and 17% kcal% sucrose; Research Diets Inc., New Brunswick, NJ, USA, D12451), while the control group of mice was kept on normal diet (ND; 13% kcal% fat and 4.6% kcal% sucrose; Special Diets Services, Witham, UK, VRF1 (P)) for the entire period. Bone marrow transplanted (BMT) mice were maintained in a specific-pathogen-free status (autoclaved top filter cages) for the entire course of the experiment, and antibiotics (amoxicillin (500 mg/L) and clavulanic acid (125 mg/L) of drinking water) were administered via the drinking water in the four weeks post transplantation. In the first two weeks of being on an ND after BMT, they were also divided into ND and HFD groups and were fed on these diets throughout the following 18 weeks. The size of the control and treated groups was selected to be similar to those reported previously in similar experiments [33]. Mice were maintained in a 12 h light/12 h darkness cycle and had access to food and water ad libitum. The body weights and food intake of the animals were registered weekly. For tissue collection, mice were killed using an isoflurane overdose at week 19 [8]. Study protocols were approved by the Animal Care and Use Committee of the University of Debrecen, under permission number 7/2021/DEMÁB.

### 2.3. Bone Marrow Transplantation

Recipient Mer-null and Mer^+/+^ wild mice (7 weeks old, males) were irradiated with 11 Gy using a Theratron 780C cobalt unit for the ablation of the recipient bone marrow. The animals to be irradiated were immobilized using a circular cage (mouse pie cage) that could hold up to 11 mice. Following irradiation, isolated bone marrow cells (in sterile RPMI-1640 medium) flushed out of the femurs, tibias, and humeri of donor Mer^−/−^ or Mer^+/+^ mice were transplanted into the recipient mice by retro-orbital injection (20 × 10^6^ bone marrow cells per mouse). This experimental protocol in our hand results in higher than 95% of repopulation [8,34].

### 2.4. Collection of Tissue Samples

At the end of the feeding phase, the animals were sacrificed with isoflurane, and body weight, body length, liver, and gonadal white adipose tissue (gWAT) weights were measured. Liver, interscapular brown adipose tissue (BAT), and gWAT samples were collected for subsequent analysis. For histological analysis, adipose tissue and liver samples were fixed in 4% paraformaldehyde; for gene expression and liver triglyceride determination were frozen in liquid nitrogen and stored at −80 °C prior to extraction [8].

### 2.5. Adipocyte and Adipose Tissue Macrophage (ATM) Isolation

gWAT tissue was dissected, washed, and kept in transport buffer (DMEM supplemented with 1% penicillin–streptomycin solution and 1% bovine serum albumin (BSA). The tissue was minced, digested for 30–60 min at 37 °C in digestion solution (HEPES buffer (pH 7.4) supplemented with 20 g/L BSA and 0.5 g/L collagenase type 1, and filtered through a nylon filter (100 μm)). After a centrifugation step, adipocytes were collected and stored at −80 °C prior to RNA isolation. After the removal of red blood cells by hemolysis (ACK Lysing Buffer, Thermo Fischer Scientific, Waltham, MA, USA), the stromal vascular cell fraction (SVC) was resuspended in staining buffer (PBS supplemented with 0.5% BSA and 2 mM EDTA). ATMs were isolated from SVCs using MACS technology (F4/80-based positive selection; Miltenyi Biotec, Gladbach, Germany, catalog numbers 130-097-050 and 130-117-509 for the beads and antibody, respectively), according to the manufacturer’s instructions [8].

### 2.6. Liver Triglyceride Levels

Triglyceride concentrations were quantified in saponified, neutralized liver extract (digested in ethanolic KOH (P5958)) overnight at 55 °C by glycerol enzymatic assay (Free Glycerol Reagent, Glycerol Standard Solution, F6428, G7793), according to the manufacturer’s instructions [8].

### 2.7. Histology

Hepatic and adipose tissues were fixed in 4% neutral buffered formaldehyde (F8775) and embedded in paraffin. Paraffin sections were stained with hematoxylin–eosin (H&E; HT110116, GHS116) stain [8]. Histological sections were analyzed on EVOS™ XL Core Imaging System (Thermo Fisher Scientific, Waltham, MA, USA). Adipose tissue images were analyzed using ImageJ software 2.0.0, (National Institutes of Health, Bethesda, MD, USA) with an Adiposoft plugin 1.16. The cross-sectional areas of adipocytes are reported in μm^2^.

### 2.8. Quantitative Real-Time Polymerase Chain Reaction (qRT-PCR) Analysis of mRNA Expression

Total RNA was isolated from various cells using the TRI reagent, according to the manufacturer’s guidelines (Thermo Fisher Scientific). Total RNA was reverse transcribed into cDNA using a High-Capacity cDNA Reverse Transcription Kit (Life Technologies, Budapest, Hungary), according to the manufacturer’s instruction. qRT-PCR was carried out in triplicate using pre-designed FAM-labeled MGB assays (Life Technologies, Budapest, Hungary) on a Roche LightCycler LC 480 real-time PCR instrument. Relative mRNA levels were calculated using the comparative CT method and were normalized to β-actin mRNA [8]. Catalog numbers of the TaqMan assays used were the following: CCL2/MCP-1 Mm00441242_m1, Tnf Mm00443258_m1, IL1B Mm00434228_m1, IL6 Mm00446190_m1, cox8 Mm00432648_m1, HSL Mm00495359_m1, ATGL Mm00503040_m1, DGAT1 Mm00499536_m1, PGC1alpha Mm01208835_m1, UCP1 Mm01244861_m1, PPARgamma Mm00440940_m1, CD36 Mm00432403_m1, and Beta-actin Mm02619580_g1.

### 2.9. Insulin Resistance Test and Serum Insulin Determination

The insulin resistance test was performed on week 17. After a 6 h fasting period, 0.75 IU/bwkg insulin (ACTRAPID Penfill 100 IU/mL, Novo Nordisk, Copenhagen, Denmark) was injected intraperitoneally. Blood glucose levels were determined with DCont Trend glucose monitor (DCont, Budapest, Hungary) at the indicated time points after insulin injection. Serum insulin levels were determined by the Mouse Insulin ELISA 80-INSMS-E01 kit (ALPCO, Salem, NH, USA), according to the manufacturer’s instructions [8].

### 2.10. Cell Culture and Differentiation of 3T3-L1 Cells

3T3-L1 murine preadipocytes (ATCC:CL-173) were maintained as subconfluent cultures in Dulbecco’s modified Eagle’s medium supplemented with 4.5 g/L D-glucose, 2 mM L-glutamine, and 10% (*v*/*v*) bovine calf serum. For inducing differentiation in T75 flasks, after two days, post-confluency 10 μg/mL insulin, 0.5 mM isobutylmethylxanthine, 2 μM rosiglitazone, and 1 μM dexamethasone were administered. The cells were maintained in this medium for 2 days and then for 3 more days in medium supplemented with 10 μg/mL insulin. The cells were then collected, seeded in 6-well plates in DMEM supplemented with 10% FCS and 10 μg/mL insulin, and divided into two groups. The control groups were exposed to 10% bovine serum albumin (BSA) alone or together with 10 nM inhibitory anti-mouse Mer antibody (Cat# 14-5751-82, Thermo Fischer Scientific) or its isotype control (Cat# 14-4321-82, Thermo Fischer Scientific), while the fatty acid-treated group was cultured in the presence of BSA-conjugated palmitate and oleate (150 or 300 μM each, prepared as it was described [35]) alone or together with 10 nM inhibitory anti-mouse Mer antibody or its isotype control. In other experiments, cells were exposed to 500 μM dibutyryl cyclic adenosine monophosphate (dbcAMP) or 10 μM forskolin as well. Cells were collected in order to detect UCP1 mRNA expressions 8 h later.

To examine lipid droplet formation, differentiated 3T3-L1 murine adipocytes were seeded in 96-well Cell Carrier Ultra plates (6055302, Perkin Elmer, Waltham, MA, USA) in DMEM supplemented with 4.5 g/L D-glucose, 2 mM L-glutamine, and 10% (*v*/*v*) bovine calf serum. The control group was exposed to BSA, while the fatty acid-treated group was cultured in the presence of BSA-conjugated palmitate and oleate alone, or together with the inhibitory anti-mouse Mer antibody or its isotype control, as described above, but for 3 days. To visualize the lipid droplets, cells were stained with HCS LipidTOX™ Deep Red Neutral Lipid Stain in 500× dilution (H34477, Thermo Fischer Scientific).

### 2.11. Examination of Lipid Droplet Formation in 3T3-L1 Cells

Images were generated with the Opera Phenix High Content Confocal System (Perkin Elmer, Waltham, MA, USA). A total of 27 fields with 600–1300 cells were acquired per well and laser-based autofocus was performed for each imaging position. Images of bright-field and Alexa-647 channels were collected at the 5 µm position of the Z image plane relative to the bottom of the optical plate using a 10× objective (NA: 1.15). In order to visualize the cells and the localization of lipid droplets, an excitation wavelength of 561 nm was used to detect fluorescence, while emissions were collected with a properly selected wavelength filter for mCherry with BP 570-630. The cells were kept at 37 °C, in a 5% CO_2_ concentration environment and the constant humidity was ensured by the special design of the Cell Carrier Ultra 96 well plate. Image analysis and the quantification of lipid droplets were performed using Harmony 4.9 software (Perkin Elmer, Waltham, MA, USA), as follows. The cell bodies were identified by detection in the bright-field channel, preceded by the contrast enhancement of the images using the smoothing method. Lipid droplet size and number were determined by identifying LipidTOX™ Deep Red-positive stained spots in the cells. We then determined the width-to-length ratio of LipidTOX™ Deep Red-positive spots and considered those with a value greater than 0.55 as a true lipid droplet. To classify lipid droplets by size, we determined their area in µm^2^ and then divided them into large (large > 700 µm^2^) and small (300 µm^2^ < small < 700 µm^2^) lipid droplets. After the evaluation of the samples, the number of lipid droplets per cell and the ratio of large/small lipid droplets in each treatment group were plotted.

### 2.12. Determination of Intracellular Cyclic Adenosine Monophosphate (cAMP) Concentrations

For the assessment of intracellular cAMP levels, differentiated 3T3-L1 murine adipocytes (7 × 10^6^) were seeded in 6-well plates in DMEM supplemented with 10% FCS and 10 μg/mL insulin as described above. The cells were kept alone or treated with 10 nM inhibitory anti-mouse Mer antibody or with 10 μM pan-TAM kinase inhibitor BMS 777607 (Cat# HY-12076, MCE, Monmouth Junction, NJ, USA) alone or with 10 μM forskolin, an adenylate cyclase activator (sc-3562; Santa Cruz Biotechnologies, Dallas, TX, USA). Cells were then resuspended in 0.1 M HCl, and the total lysate was extracted. The concentration of intracellular cAMP was determined with a cAMP ELISA kit (Enzo Life Sciences, Farmingdale, NY, USA), according to the manufacturer’s instructions.

### 2.13. Statistical Analysis

Data are presented as either mean ± SD or mean and individual values as indicated. All statistical analyses were performed using GraphPad Prism 6.01. For differences between four or six groups one-way ANOVA (Sidak’s multiple comparison test) was used. For datasets split on two independent factors, two-way ANOVA (multiple comparison procedures with Tukey post hoc test) was used. * denotes *p* ≤ 0.05, ** denotes *p* ≤ 0.01, *** denotes *p* ≤ 0.001, and **** denotes *p* ≤ 0.0001.

## 3. Results

### 3.1. Mer-Null Mice Respond to High-Fat Diet with Attenuated Body Weight Gain

As compared with animals kept on a standard control diet, Mer^+/+^ mice exposed to HFD developed marked obesity (38.6 ± 2.0 vs. 29.6 ± 1.5 g body weight of mice kept on HFD or ND, respectively, *p* < 0.0001). Loss of Mer did not affect significantly the body weight of mice at the start of the feeding experiment (19.0 ± 0.6 vs. 18.1 ± 2.4 g body weight of Mer^+/+^ vs. Mer^−/−^ mice, respectively *p* < 0.35). However, in the longer term, Mer-null mice seemed to exhibit a slower body weight gain concluded from the body weights of the ND eating groups at the end of the feeding experiment (29.6 ± 1.5 vs. 27.0 ± 0.9 g body weight of Mer^+/+^ vs. Mer^−/−^ mice, respectively; *p* < 0.043). The decreased ability of Mer-null mice to gain body weight as compared to their wild-type littermates was more pronounced when the mice were kept on HFD (Figure 1A–C).

In accordance with the body weight data, the gonadal fat tissue (gWAT) weight of Mer^−/−^ mice was also significantly lower than that of the Mer^+/+^ mice kept on ND (0.63 ± 0.08 vs. 0.24 ± 0.02 g gWAT weights of Mer^+/+^ vs. Mer^−/−^ mice, respectively; *p* < 0.05). In addition, while it increased significantly in Mer^+/+^ mice exposed to HFD, no significant alterations were found in the case of Mer^−/−^ mice under similar feeding conditions (2.19 ± 0.12 vs. 0.49 ± 0.06 g gWAT weights of Mer^+/+^ vs. Mer^−/−^ mice exposed to HFD, respectively; *p* < 0.0001) (Figure 1D). The average size of adipocytes on the histology sections reflected the gWAT weight changes observed (Figure 1E,F).

Since previous reports indicated that the hypertrophy of adipocytes is associated with enhanced pro-inflammatory cytokine and altered adipokine production [4,11,12], we decided to determine the mRNA expressions of these signaling molecules in the gWAT. As seen in Figure 1G, the mRNA expression levels of pro-inflammatory cytokines, such as MCP-1, TNF-α, and interleukin (IL)-6 were increased in the gWAT adipocytes of wild-type mice exposed to HFD. However, no significant increase was detected in the levels of pro-inflammatory cytokines of the gWAT adipocytes of HFD-exposed Mer-null mice. Similarly, resistin and leptin mRNA levels were significantly enhanced, while adiponectin mRNA levels were significantly decreased in the gWAT adipocytes of HFD-exposed wild-type mice, while no such changes were detected in their HFD-exposed Mer-null littermates.

### 3.2. Loss of Mer in Mice Protects Against High-Fat Diet-Induced Hepatosteatosis and Insulin Resistance

HFD significantly enhanced the mRNA expressions of several pro-inflammatory cytokines, as well as that of resistin in the Mer^+/+^ ATMs, while Mer mRNA expressions were significantly lower. In contrast, significantly fewer or no alterations were seen in Mer^−/−^ MΦs exposed to HFD (Figure 2A). mRNA expression alterations in the wild-type ATMs reflect the change in the composition of MΦs in obesity: in non-obese mice, the ATMs are tissue-resident MΦs known to express high levels of Mer but low levels of pro-inflammatory cytokines, while in HFD mice, pro-inflammatory MΦs infiltrate the adipose tissue and express low levels of Mer [26]. Altogether, these data indicate that Mer-null adipocytes do not become hypertrophic when Mer^−/−^ mice are exposed to long-term HFD, and consequently, inflammation does not develop in the HFD-exposed Mer-null adipose tissue.

Since inflammation and altered adipokine production triggered by HFD have been associated with insulin resistance and hepatosteatosis [13], we determined the development of hepatic steatosis and the alterations in insulin levels and insulin resistance in these mice. As shown in Figure 2B, no alterations were found in the liver weights of mice under the two different feeding conditions. However, in the wild-type livers exposed to HFD, a significant increase in the triacylglycerol content was detected, as compared to their ND-fed counterparts. In striking contrast, the triacylglycerol content of Mer^−/−^ livers exposed to HFD was not altered as compared to their ND-exposed Mer^−/−^ counterparts (Figure 2C). Hematoxylin–eosin-stained tissue sections of the livers confirmed the above findings (Figure 2D).

Next, the insulin resistance tests were performed, which indicated a significantly enhanced insulin resistance in the HFD-exposed wild-type mice, while no insulin resistance developed in HFD-exposed Mer^−/−^ mice (Figure 2E). In accordance with these observations, compensatory increases in the serum insulin levels of the HFD-exposed Mer^+/+^ mice were detected, while no changes in insulin levels were seen in the case of Mer-null mice (Figure 2F). Altogether, our observations indicate that loss of Mer has a protective effect against HFD-induced obesity and related pathological alterations in mice.

### 3.3. Loss of Mer Expressed by Nonhematopoietic Cells Protects Against HFD-Induced Obesity

The findings with the Mer-null mice were so different from the ones we expected that we decided to determine whether the loss of Mer from the bone marrow-derived cells or other cell types is responsible for the above phenomena. For this purpose, wild-type or Mer-null mice were terminally irradiated, and their bone marrow was replaced by bone marrow originating from either wild-type or Mer-null mice, as indicated in Figure 3. Mer^+/+^ mice transplanted with Mer^+/+^ bone marrow served as a control. Irradiated mice gained significantly less weight than their non-irradiated counterparts by the end of the 18 weeks of feeding (Figure 3A), in accordance with the observations of previous studies [36,37]. Still, body weight changes clearly demonstrated that the loss of Mer from the hematopoietic cells did not affect HFD-induced body weight gain. If, however, Mer was missing from the nonhematopoietic cells, mice were protected against the HFD-induced weight gain, at least in the early weeks of the feeding experiment (Figure 3A).

As we expected originally, loss of Mer from the hematopoietic cells led to a higher gWAT size in mice exposed to HFD, as compared to the HFD-exposed wild-type mice. The difference was even more significant if it was compared to the gWAT of those HFD-exposed mice in which Mer was missing from the nonhematopoietic cells (Figure 3B). Hematoxylin–eosin-stained tissue sections, in addition, also revealed that the basal size of adipocytes of those mice in which Mer was missing from the hematopoietic cells was higher already on ND, as compared to the other mice (Figure 3C,D).

In accordance with the weight gain data, an increase in the liver triacylglycerol content was also detected in all three types of mice exposed to HFD (Figure 3E,F), with the highest amount observed in mice with Mer missing from their hematopoietic cells, and this increase was significantly higher than in mice with Mer missing from nonhematopoietic cells. However, at this low weight gain, no significant differences in insulin sensitivity were developed (Figure 3G). Altogether, our data indicate that the loss of Mer from the nonhematopoietic cells plays the dominant role in the protection against HFD-induced obesity in mice.

### 3.4. Mer Is Expressed in White Adipocytes and Promotes Their Triacylglycerol Accumulation

Thus, instead of investigating further the role of MΦ Mer in the adipose tissue homeostasis as we originally planned, we decided to determine whether TAM kinases are expressed by adipocytes of the gWAT in mice. As seen in Figure 4A, all three tyrosine kinases are expressed on mRNA level by gWAT adipocytes, Axl, and Mer with nearly the same mRNA expression levels. Exposure to HFD in wild-type mice led to a decrease in the mRNA expression of Mer and Tyro3, while an increase in that of Axl. The loss of Mer did not affect the mRNA expressions of Tyro3, but Axl failed to increase in the HFD-exposed Mer^−/−^ gWAT.

Adipocyte enlargement results from an increase in the lipid droplet size. One possibility for the lack of adipocyte hypertrophy in the absence of Mer is that Mer tyrosine kinase activity contributes to the growth of white adipocytes, as insulin receptors do. To investigate this possibility, we followed the differentiation of fatty acid-exposed 3T3-L1 fibroblasts to adipocytes (an in vitro model of white adipocyte differentiation) [38] in the presence and absence of Mer signaling. As seen in Figure 4B, Mer was nearly absent from 3T3-L1 fibroblasts, but its mRNA expression significantly increased until day 7 of adipocyte differentiation, then it gradually decreased.

Exposure of adipocytes on day 7 of their differentiation to BSA, which carries bound fatty acids, already enhanced their lipid accumulation. However, the accumulation was more pronounced when they were exposed either to lower (150 μM) or especially to higher (300 μM) concentrations of fatty acids, resulting in a high percentage of adipocytes expressing large lipid droplets detected 3 days later (Figure 4C,D). The addition of an inhibitory Mer antibody at the time of fatty acid administration attenuated the fatty acid-induced large lipid droplet formation characteristic for white adipocytes, while its isotype control had no effect (Figure 4C,D). These data indicate that Mer signaling might contribute to the growth of adipocytes.

Since previous reports indicated that 3T3-L1 adipocytes display phenotypic characteristics of multiple adipocyte lineages [39], we decided to test whether the lack of Mer signaling promotes browning in 3T3-L1 adipocytes and whether this is how it attenuates adipocyte hypertrophy. For this purpose, we determined the alterations in the mRNA levels of the uncoupling protein-1 (UCP1), a crucial mitochondrial protein, which mediates thermogenesis, and is expressed by both brown and beige adipocytes [40]. As seen in Figure 4E, the administration of dbcAMP, a cell-permeable synthetic analog of cAMP, a known inducer of UCP1 expression [41], strongly induced the mRNA expression of UCP1 within 8 h, confirming the involvement of adenylate cyclase pathway in the regulation of UCP1 expression in the brown-like 3T3 adipocytes as well. The inhibition of Mer signaling alone or in fatty acid-exposed 3T3-L1 adipocytes, however, did not trigger a significant increase in the mRNA expression of UCP1, indicating no activation of thermogenesis in the absence of Mer signaling in 3T3-L1 adipocytes.

### 3.5. Browning Gene Expressions Are Not Altered in the Gonadal Fat Tissue of the HFD-Exposed Mer-Null Mice

As the difference in the body weight gain in the absence of Mer was not related to an altered energy intake, as Mer^+/+^ and Mer^−/−^ mice consumed the same amount of food during the feeding experiment (Figure 5A), wild-type mice must have stored the extra energy taken up in the form of triacylglycerol in the gWAT, while Mer-null mice consumed it. Since the enhanced browning activity of the white adipose tissue is known to be involved in body weight control by facilitating thermogenesis [42], we determined the mRNA expressions of genes associated with the browning process in the gWAT of HFD-exposed mice (Figure 5B). However, no significant alterations in the mRNA expressions of UCP1, or that of COX8, a subunit of cytochrome c oxidase, that would reflect enhanced mitochondrial biogenesis were detected. In addition, we also determined the mRNA levels of hormone-sensitive lipase (HSL) and adipose tissue triacylglycerol lipase (ATGL) enzymes involved in triacylglycerol degradation [43], as well as those of the diacylglycerol O-acyltransferase (DAGT1), an endoplasmic reticulum (ER) enzyme that protects ER from lipotoxicity [44] (Figure 5C). We found that the expression of lipases was significantly decreased in the HFD-exposed Mer^+/+^ gWAT, while no such alterations were detected in the Mer^−/−^ gWAT, indicating that while in Mer^+/+^ adipocytes the triacylglycerol degradation might have been attenuated following exposure to HFD, contributing to the adipocyte hypertrophy, it was not altered in the Mer^−/−^ adipocytes. At the same time, no strain differences in the HFD-induced DGAT1 mRNA expressions were found.

### 3.6. Enhanced Whitening in the Brown Adipose Tissue of HFD-Exposed Mer^+/+^ and Enhanced Thermogenesis in That of HFD-Exposed Mer^−/−^ Mice

Since gWAT adipocytes did not show enhanced browning, we looked at the response of interscapular BAT to the HFD. As shown in Figure 6A, BAT also expresses TAM kinases, with the highest mRNA expression of Axl. While the mRNA expression of Axl and Tyro3 did not alter in response to the HFD or to the loss of Mer, the mRNA expression of Mer tyrosine kinase increased significantly in the BAT of wild-type animals exposed to HFD.

Similarly to gWAT, exposure of wild-type mice to HFD led to a significant increase in the mRNA expression of the pro-inflammatory cytokine MCP-1 and TNFα in BAT adipocytes but did not affect the mRNA expression of IL-6, known to be released from browning BAT adipocytes and to promote the browning process [45] (Figure 6B). In the absence of Mer, exposure to HFD did not trigger a pro-inflammatory response in BAT adipocytes, but the mRNA expression of IL-6 significantly increased indicating a potentially increased thermogenesis. Indeed, the mRNA expression levels of genes associated with enhanced thermogenesis, such as peroxisome proliferator-activated receptor (PPAR)γ [46], and PPARγ coactivator (PGC) 1α [47], master regulators of mitochondrial biogenesis, UCP1, and that of genes associated with metabolic pathways providing sufficient amounts of fatty acid substrates for thermogenesis, such as the CD36 fatty acid transporter participating in the fatty acid uptake [48], and the lipolytic HSL and ATGL were all elevated in HFD-exposed Mer^−/−^ BAT, as compared to the HFD-exposed wild-type BAT. The ER-protective DGAT1 had also significantly elevated expression in HFD-exposed Mer^−/−^ BAT as compared to Mer^+/+^ BAT.

Histological sections of BAT tissue from wild-type mice kept on HFD revealed enhanced whitening [49,50], while no whitening was observed in the BAT of HFD-exposed Mer-null mice (Figure 6C).

Previous studies indicated that a loss of Axl signaling in adipocytes also results in enhanced thermogenesis, and in protection against diet-induced obesity. Studies conducted on brown adipocyte cell lines indicated that this is related to an attenuation of the AKT-mediated activation of cAMP phosphodiesterases in the absence of Axl signaling, leading to elevations in the intracellular cAMP levels [51]. To find out whether Mer mediates its effects on brown adipocytes also via decreasing the intracellular cAMP levels, we took advantage of the heterogeneity of the 3T3-L1 adipocyte cell line and determined its cytosolic cAMP levels if kept in medium or following exposure to anti-Mer antibodies; to the pan-TAM kinase inhibitor BMS-777607; to forskolin, an adenylate cyclase activator; or to a combination of these. As seen in Figure 7A, 3T3-L1 cells tested on the 7th day of their differentiation express all three TAM kinases, with Axl expression being the highest. The inhibition of Mer, or all the TAM kinases alone, only slightly increased the basal intracellular cAMP levels, as not much signal capable of maintaining a high adenylate cyclase activity was present in the cell culture medium. However, a significant increase in cAMP levels induced by inhibition of Mer or all the TAM kinases was detected, when adenylate cyclase was simultaneously triggered by forskolin (Figure 7B), confirming previous findings [51]. Increases in cAMP levels were followed by induction of the UCP1 mRNA expressions (Figure 7C).

## 4. Discussion

In the presented experiments, the involvement of Mer receptor tyrosine kinase was investigated in the maintenance of adipose tissue homeostasis. Since MΦs are known to contribute to adipose tissue homeostasis and Mer is highly expressed by tissue-resident MΦs, our prediction was that Mer acting in the myeloid compartment would have a protective role against HFD-induced adiposity and inflammation. Indeed, bone marrow transplantation experiments in which Mer was missing from the hematopoietic cells indicated enhanced adipocyte hypertrophy in mice kept on both ND and HFD. The limitation of this study is the strong negative influence of irradiation on the ability of mice to gain body weight. Thus, to address the precise role of Mer in the adipose tissue homeostasis played in the myeloid compartment, mice carrying the myeloid-specific deletion of Mer should be investigated in the future.

Unexpectedly, Mer is also expressed by adipocytes, and the simultaneous loss of Mer in adipocytes of the full-body knock-out mice overwrote the myeloid-specific Mer-null phenotype. In the absence of Mer, not only is the hypertrophy of white adipocytes was attenuated but Mer has such a strong inhibitory effect on the thermogenesis in the BAT that its loss led to a sufficient increase in the rate of thermogenesis to prevent lipid accumulation in the Mer-null mice kept on HFD diet. Our data indicate that the enhanced thermogenesis of brown adipocytes in the absence of Mer might be related to the regulation of intracellular cAMP levels by Mer, similar to the other TAM kinase Axl, which was reported to inhibit cAMP phosphodiesterases in brown adipocytes via the activation of AKT [51]. Similarly to Axl, Mer is also known to activate AKT [52].

The lack of white adipocyte hypertrophy in Mer-null mice exposed to HFD could be solely due to the enhanced thermogenesis detected in the BAT. Our results, conducted on the 3T3-L1 adipocyte cell line, however, indicate that Mer can also directly affect the hypertrophy of adipocytes exposed to fatty acids. The signaling pathway via which Mer promotes the hypertrophy of adipocytes was not investigated in our study. However, mTOR, especially mTORC1, has been implicated to play a crucial role in promoting adipocyte growth and fat accumulation in the signaling pathway induced by two other receptor tyrosine kinases, insulin and insulin-like growth factor receptor 1 (IGFR1) [3,53]. The data obtained here also indicate that the Mer-induced adipocyte hypertrophy pathway is not WAT specific, as it also promotes lipid accumulation in the BAT adipocytes of wild-type mice exposed to HFD, leading to the whitening of the BAT tissue [49,50]. Fat accumulation in the BAT, similar to WAT, triggers pro-inflammatory macrophage recruitment, and the pro-inflammatory cytokines released might also contribute to the inhibition of thermogenesis [54].

The fact that the intensity of the tyrosine kinase signaling pathways in adipocytes plays a decision role between storing excess lipids or converting them to heat has already been indicated by previous studies. Thus, the injection of a single dose of insulin decreased the expression of PGC1α and UCP1 in BAT [55]. Accordingly, mice generated by the deletion of one of the insulin alleles resulting in lower diet-induced insulin levels were resistant to diet-induced obesity and showed increased browning of WAT [56]. Interestingly, not only the tyrosine kinase pathway triggered by insulin might contribute to this metabolic decision, since mice characterized by adipocyte-specific deletion of IGFR1 [57], or of Axl [51] showed also protection against HFD-induced obesity and enhanced BAT thermogenesis in mice. Our experimental data add Mer to the list of tyrosine kinase receptors that promote adipocyte hypertrophy, while negatively regulating adipocyte thermogenesis, and indicate that adipocyte-specific targeting of Mer might provide a potential tool in regulating obesity. Since, however, the loss of Gas6, a ligand for both Mer and Axl, also protects mice against HFD-induced obesity [58], and obesity in humans is associated with increased plasma Gas6 levels [59,60], decreasing Gas6 levels by neutralizing antibodies [61] might be a more realistic approach to target obesity.

## 5. Conclusions

Our present research and previous studies published by others indicate that enhanced tyrosine kinase activity in adipocytes promotes adipocyte hypertrophy and inhibits thermogenesis in HFD-induced mice favoring lipid storage. Interestingly, all the ligands for the tyrosine kinase receptors expressed by adipocytes, such as IGF-1, Protein S, Gas6, or growth differentiation factor 3, which also promotes adipocyte growth [62], are known to be produced by engulfing macrophages, and their loss is associated with a decreased sensitivity to HFD-induced obesity. Thus, our findings reveal a novel interaction between adipocytes and adipose tissue macrophages in the control of body weight.

## Figures and Tables

**Figure 1 cells-13-01902-f001:**
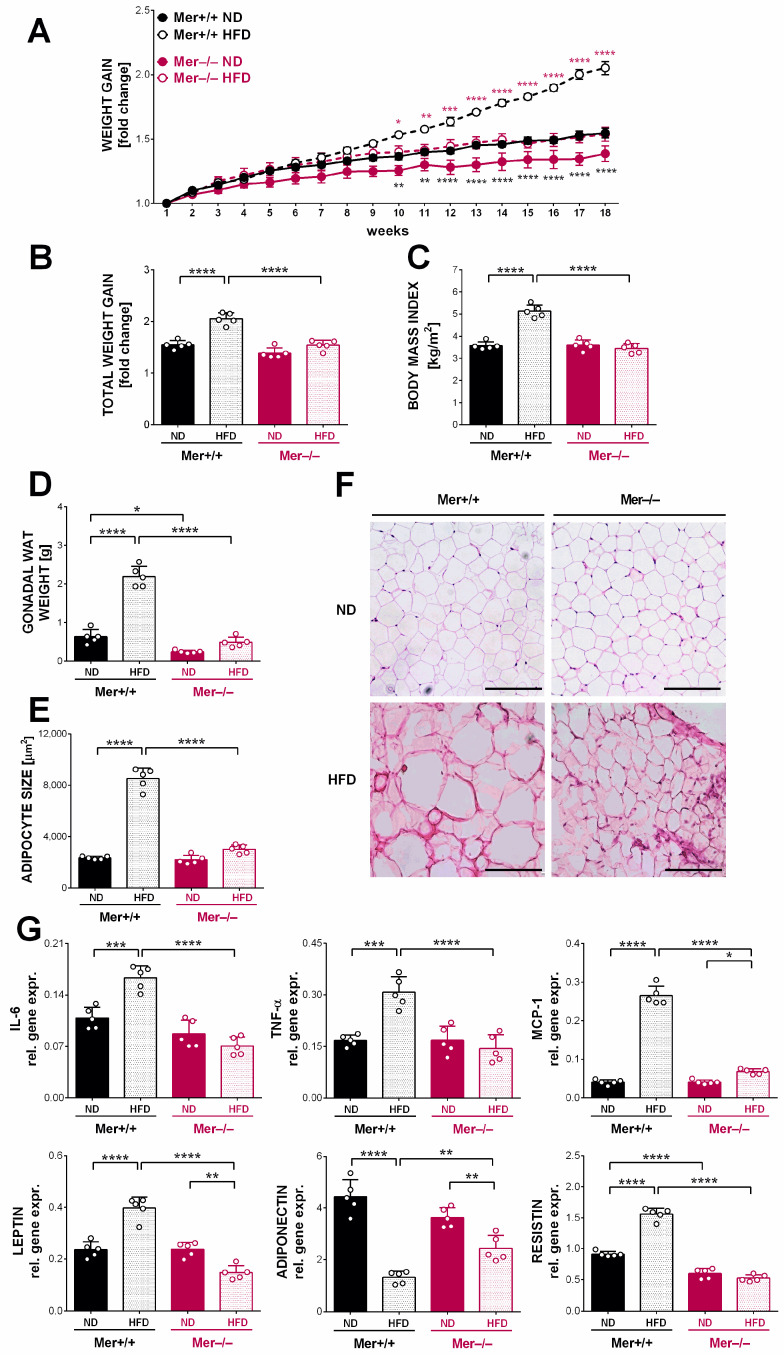
Mer-null mice are protected against diet-induced obesity. (**A**) Weekly body weight gain of Mer^+/+^ and Mer^−/−^ mice kept on either ND or HFD. Black stars indicate statistically different values between the wild-type ND and HFD groups, while pink stars indicate the difference between the two HFD groups on the same day. (**B**) Total weight gain of Mer^+/+^ and Mer^−/−^ mice kept on either ND or HFD at the end of the 18-week feeding experiment. Data in (**A**,**B**) are expressed as fold change as compared to the body weight determined at the beginning of the feeding experiment. (**C**) Body mass index of Mer^+/+^ and Mer^−/−^ mice kept on either ND or HFD determined at the end of the feeding experiment. (**D**) Gonadal WAT weight of Mer^+/+^ and Mer^−/−^ mice kept on either ND or HFD at the end of the feeding experiment. (**E**,**F**) Paraffin-embedded gonadal WAT slides from Mer^+/+^ and Mer^−/−^ mice kept on either ND or HFD stained with H&E to visualize tissue architecture at the end of the feeding experiment together with the means of adipocyte sizes determined from the slides. ImageJ software was used to examine 500 or more adipocytes in each sample. One representative series of five is shown. Scale bar, 200 μm. (**G**) Relative gene expression levels of several adipokines and pro-inflammatory cytokines in gWAT adipocytes from Mer^+/+^ and Mer^−/−^ mice kept on either ND or HFD were determined by qRT-PCR at the end of the feeding experiment. All the data are presented as mean and individual values, and statistical significance was evaluated by two-way ANOVA. * denotes *p* ≤ 0.05, ** denotes *p* ≤ 0.01, *** denotes *p* ≤ 0.001, and **** denotes *p* ≤ 0.0001.

**Figure 2 cells-13-01902-f002:**
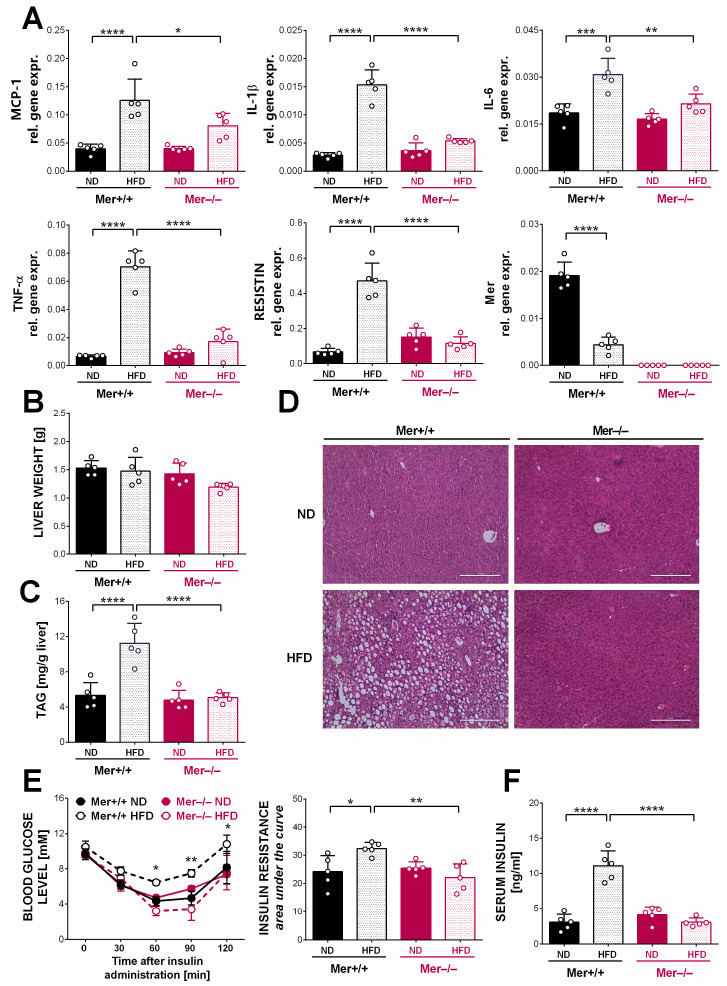
Loss of Mer protects mice against HFD-induced inflammation, hepatosteatosis, and insulin resistance. (**A**) Relative gene expression levels of several pro-inflammatory cytokines, resistin, and Mer in gWAT macrophages from Mer^+/+^ and Mer^−/−^ mice kept on either ND or HFD determined by qRT-PCR at the end of the feeding experiment. (**B**) Liver weights of Mer^+/+^ and Mer^−/−^ mice kept on either ND or HFD determined at the end of the feeding experiment. (**C**) Triacylglycerol content of livers from Mer^+/+^ and Mer^−/−^ mice kept on either ND or HFD determined at the end of the feeding experiment. (**D**) Paraffin-embedded hepatic tissue slides stained with H&E to visualize tissue architecture at the end of the feeding experiment. One representative series of three is shown. Scale bar, 200 μm. (**E**) Insulin resistance values of Mer^+/+^ and Mer^−/−^ mice kept on either ND or HFD determined on week 17 of the feeding experiment. (**F**) Serum insulin levels of Mer^+/+^ and Mer^−/−^ mice kept on either ND or HFD determined at the end of the feeding experiment. All the data are presented as mean and individual values, and statistical significance was evaluated by two-way ANOVA. * denotes *p* ≤ 0.05, ** denotes *p* ≤ 0.01, *** denotes *p* ≤ 0.001, and **** denotes *p* ≤ 0.0001.

**Figure 3 cells-13-01902-f003:**
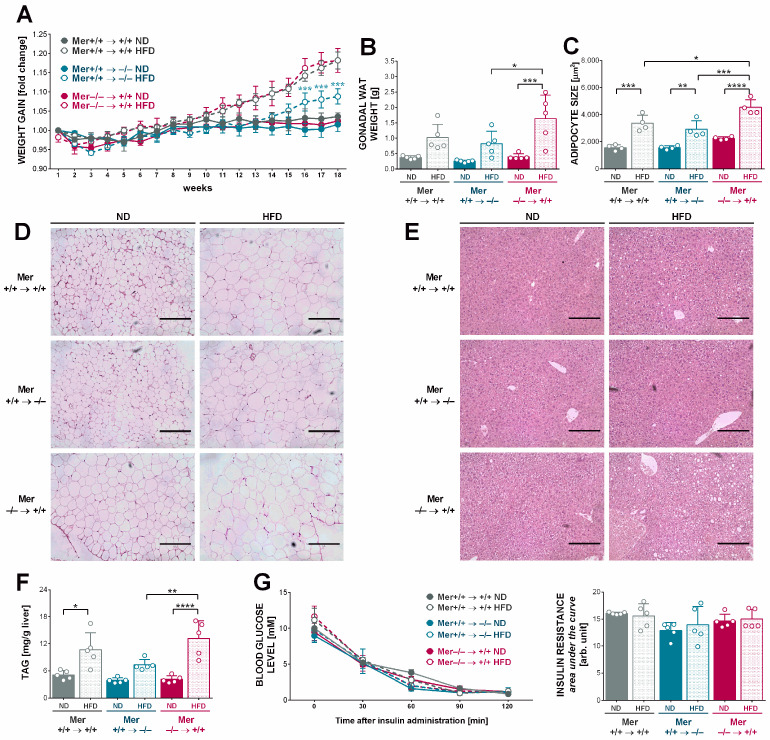
Loss of Mer from the nonhematopoietic cells protects mice against HFD-induced obesity. (**A**) Weekly body weight gain of bone marrow-transplanted mice kept on either ND or HFD. Blue stars indicate statistically different values between HFD-exposed wild-type mice as compared to those HFD-exposed mice in which Mer was missing from the nonhematopoietic cells. Arrows indicate which type of bone marrow cells are transplanted into which genetic background type of host animals. (**B**) gWAT mass of bone marrow transplanted mice kept on either ND or HFD determined at the end of the feeding experiment. (**C**,**D**) Paraffin-embedded gWAT slides from various mice stained with H&E to visualize tissue architecture at the end of the feeding experiment together with the means of adipocyte sizes determined from the slides. ImageJ software was used to examine 500 or more adipocytes in each sample. One representative series of four is shown. Scale bar, 200 μm. (**E**) Paraffin-embedded hepatic tissue slides from various mice samples stained with H&E to visualize tissue architecture at the end of the feeding experiment. One representative series of three is shown. Scale bar, 200 μm. (**F**) Triacylglycerol content of livers from bone marrow-transplanted mice kept on either ND or HFD at the end of the feeding experiment. (**G**) Insulin resistance values of bone marrow transplanted mice kept on either ND or HFD determined on week 17 of the feeding experiment. All the data are presented as mean and individual values, and statistical significance was evaluated by two-way ANOVA. * denotes *p* ≤ 0.05, ** denotes *p* ≤ 0.01, *** denotes *p* ≤ 0.001, and **** denotes *p* ≤ 0.0001.

**Figure 4 cells-13-01902-f004:**
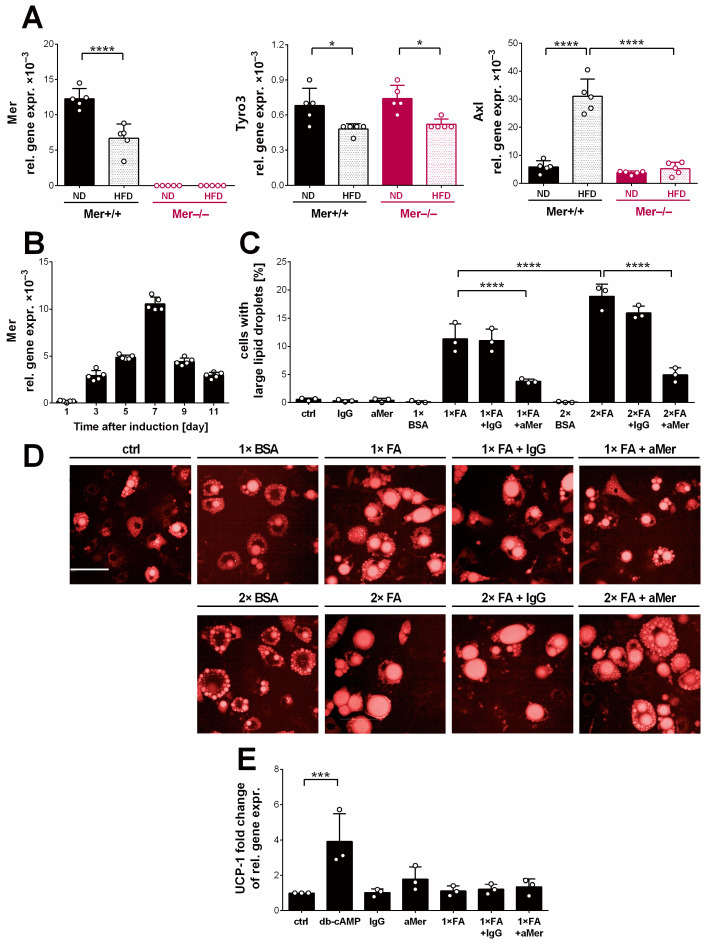
TAM kinases are expressed in white adipocytes. (**A**) Relative gene expression levels of TAM kinases in the gWAT adipocytes of Mer^+/+^ and Mer^−/−^ mice exposed to either ND or HFD determined by qRT-PCR at the end of the feeding experiment. (**B**) Changes in the mRNA expression levels of Mer during the differentiation of 3T3-L1 adipocytes. (**C**) Percentage and (**D**) images of 3T3-L1 adipocytes with large lipid droplets, after exposure for 3 days to BSA, to palmitate/oleate (150 μM each; 1× FA), or to (300 μM each; 2× FA) alone or in the presence of an inhibitory anti-Mer antibody or its isotype control. Scale bar, 100 μm. (**E**) Relative UCP1 mRNA expression in 3T3 adipocytes after 8 h exposure to 500 μM dbcAMP, to BSA, or to 1× FA alone or in the presence of the anti-Mer antibody or its isotype control. All the data are presented as mean and individual values, and statistical significance was evaluated by one-way ANOVA, except in A, where it was evaluated by two-way ANOVA. * denotes *p* ≤ 0.05, *** denotes *p* ≤ 0.001, and **** denotes *p* ≤ 0.0001.

**Figure 5 cells-13-01902-f005:**
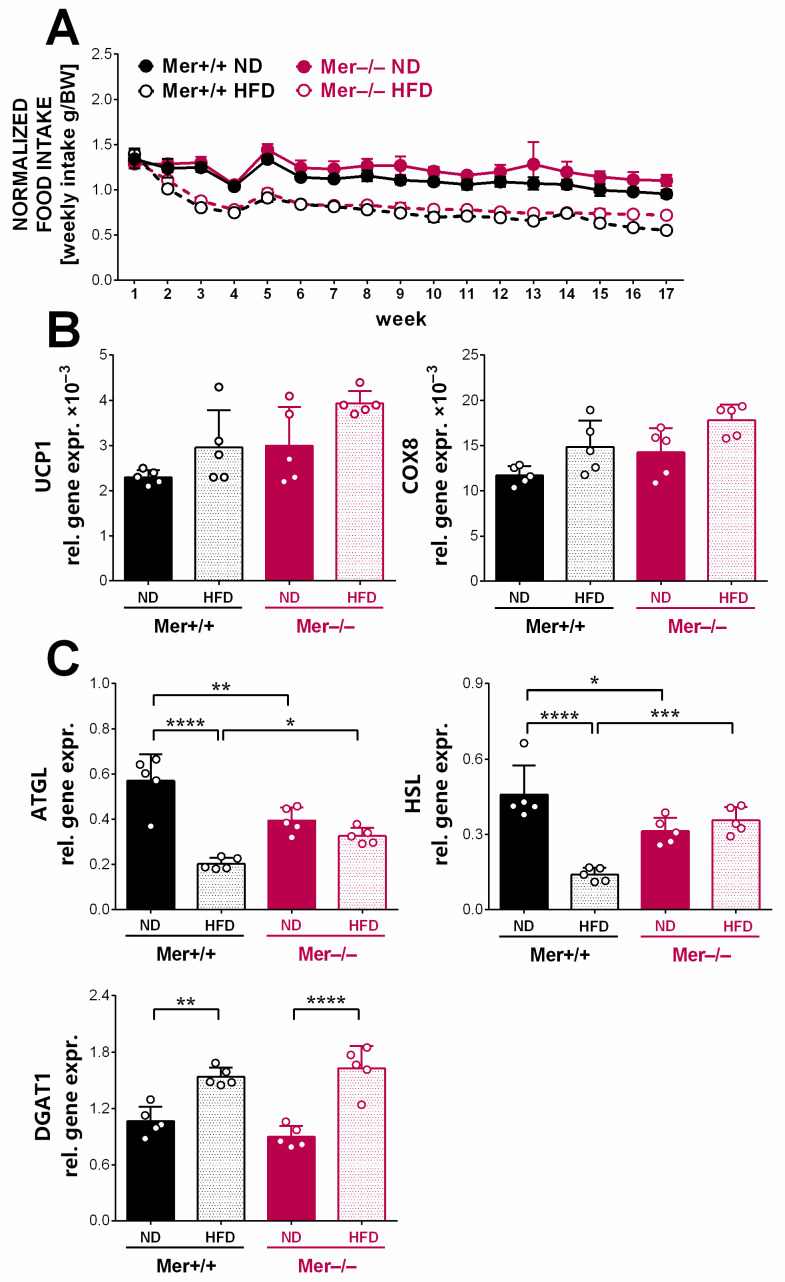
Genes related to browning are not induced in the gWAT of HFD-exposed Mer^−/−^ mice. (**A**) Weekly food intake of Mer^+/+^ and Mer^−/−^ mice exposed to either ND or HFD. (**B**) Relative gene expression levels of several browning-related and (**C**) lipid metabolic genes in the gonadal fat tissue of Mer^+/+^ and Mer^−/−^ mice exposed to either ND or HFD determined by qRT-PCR at the end of the feeding experiment. All the data are presented as mean and individual values, and statistical significance was evaluated by two-way ANOVA. * denotes *p* ≤ 0.05, ** denotes *p* ≤ 0.01, *** denotes *p* ≤ 0.001, and **** denotes *p* ≤ 0.0001.

**Figure 6 cells-13-01902-f006:**
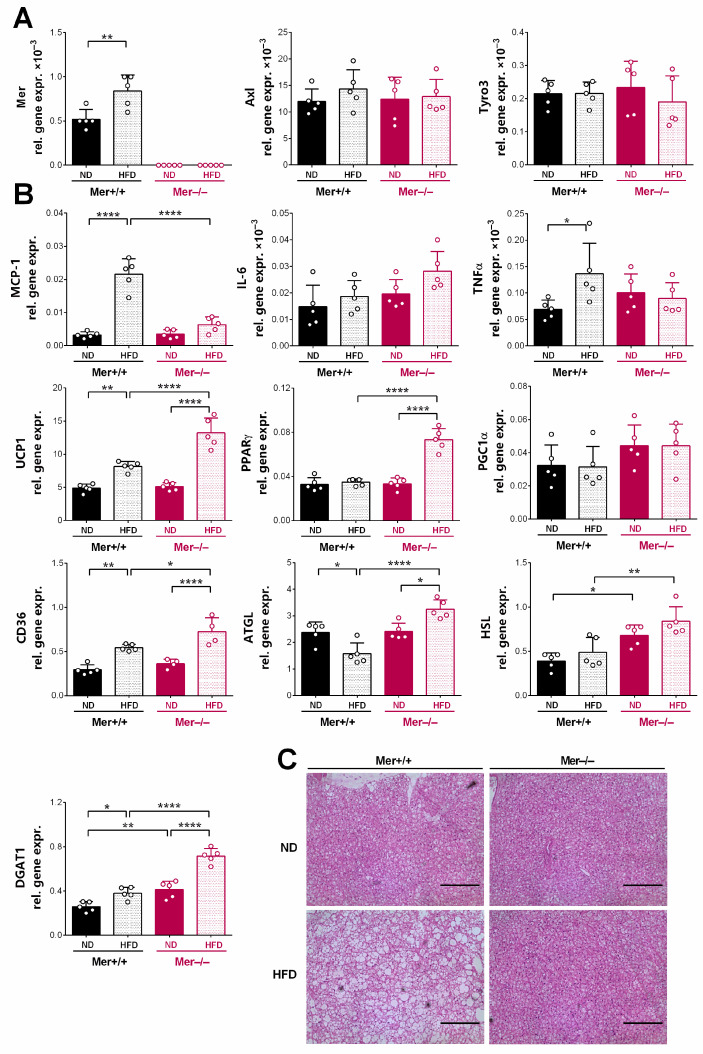
Exposure to HFD leads to the whitening of the BAT in Mer^+/+^ and to enhanced thermogenesis in the BAT of Mer^−/−^ mice, respectively. (**A**) Relative gene expression levels of TAM kinases in the interscapular BAT of Mer^+/+^ and Mer^−/−^ mice exposed to either ND or HFD determined by qRT-PCR at the end of the feeding experiment. (**B**) Relative gene expression levels of several browning-related and pro-inflammatory genes in the interscapular BAT of Mer^+/+^ and Mer^−/−^ mice exposed to either ND or HFD determined by qRT-PCR at the end of the feeding experiment. (**C**) Paraffin-embedded interscapular BAT slides of Mer^+/+^ and Mer^−/−^ mice exposed to either ND or HFD stained with H&E to visualize tissue architecture at the end of the feeding experiment. Scale bar, 200 μm. All data are presented as mean and individual values, and statistical significance was evaluated by two-way ANOVA. * denotes *p* ≤ 0.05, ** denotes *p* ≤ 0.01, and **** denotes *p* ≤ 0.0001.

**Figure 7 cells-13-01902-f007:**
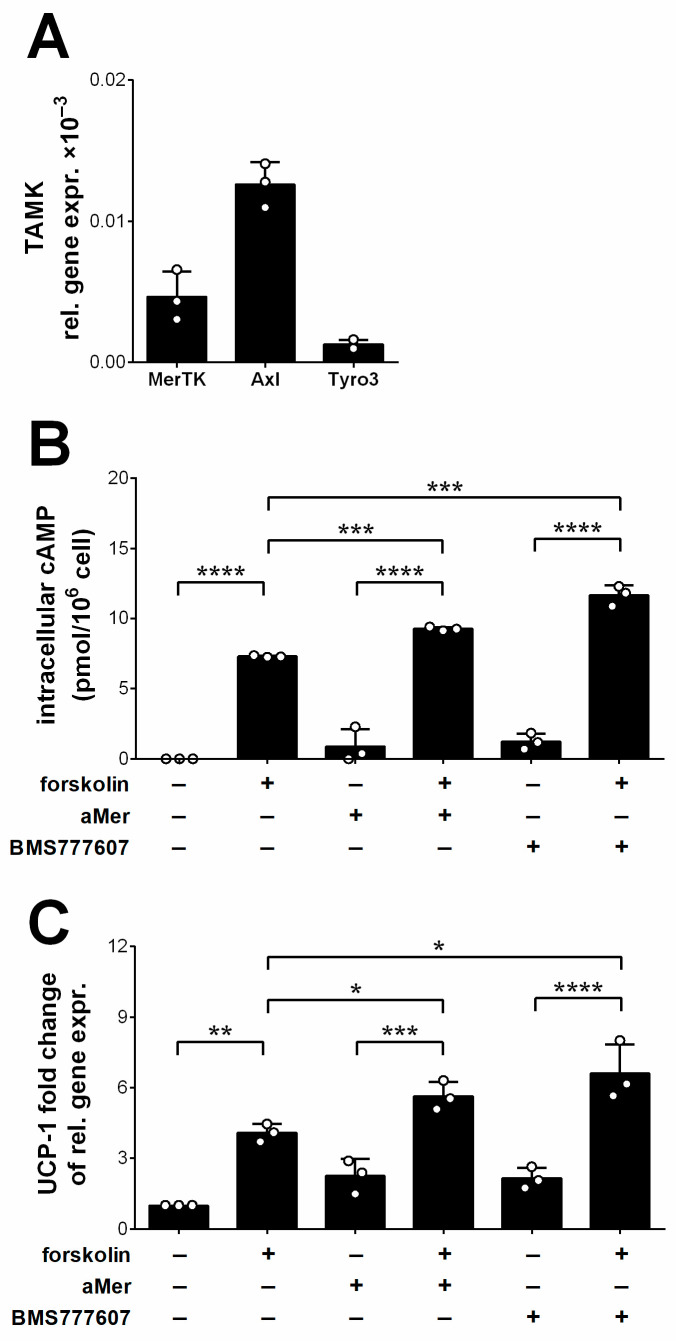
Decreased TAM kinase activity leads to enhanced intracellular cAMP levels in the presence of forskolin, an adenylate cyclase activator. (**A**) Relative gene expressions of TAM kinases in the 3T3-L1 adipocytes determined on the 7th day of their differentiation determined by RT-qPCR. (**B**) Changes in the intracellular cAMP levels of 3T3-L1 adipocytes following the administration of the anti-Mer antibody or BMS-777607, a pan-TAM kinase inhibitor, in the presence and absence of forskolin, an adenylate cyclase activator. The aMer antibody and BMS777607 were added 30 min before the addition of forskolin and cAMP levels were determined 10 min after the administration of forskolin. (**C**) Changes in the mRNA levels of UCP1 of 3T3-L1 adipocytes following the administration of the anti-Mer antibody or BMS-777607, a pan-TAM kinase inhibitor, in the presence and absence of forskolin, an adenylate cyclase activator determined by qRT-PCR. As in the previous experiment, the antibody and the TAM kinase inhibitor were added 30 min prior to forskolin, and qRT-PCR measurements were performed after 8 h of the administration of the forskolin. All data are presented as mean and individual values, and statistical significance was evaluated by two-way ANOVA. * denotes *p* ≤ 0.05, ** denotes *p* ≤ 0.01, *** denotes *p* ≤ 0.001, and **** denotes *p* ≤ 0.0001.

## Data Availability

Data are contained within the article.

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
