# Peer review of "Loss of MER Tyrosine Kinase Attenuates Adipocyte Hypertrophy and Leads to Enhanced Thermogenesis in Mice Exposed to High-Fat Diet"

_cells, 2024, doi:10.3390/cells13221902_

Round 1

Reviewer 1 Report

Comments and Suggestions for Authors

Authors present an interesting work about the effect of MER tyrosine kinase on adipocyte hypertrophy and thermogenesis. They describe a novel interaction useful to understand adiposse tissue metabolism.

There are some aspects I think they may clarify and improve:

1. The last sentence of the introduction (lines 89-91) may be omitted.  It adds nothing new and raises doubts about the results obtained in the work. 

2. In results, lines 266-267. Since the values ​​of weight changes have been described in body weight, the numerical value should also be express in the change in weight of gonadal fat tissue.

3. Figure 2a shows that HFD significantly inhibits Mer expression in Mer +/+ gWAT macrophages and the figure caption is Mer loss protects mice against HFD-induced inflammation, hepatosteatosis and insulin resistance. We know that HFD induce inflammation in adipose tissue. How does authors explain this data?

4. Figures 2b and 2c show liver weights and liver TAG content. In Mer +/+ mice HFD significantly increases TAGs but liver weights do not change. How does authors explain this data?

5. In figure 3 should order the letters of the graphics

6. In section 3.4 authors evaluate the expression of TAM kinases. Authors should explain better the importance of these kinases

Comments on the Quality of English Language

Authors should review the quality of the English and edit some phrases and expressions to improve the text.

Author Response

We would like to thank the reviewers for their careful work with which they strongly increased the clarity and the value of our paper, and we hope that they can accept our paper in the revised version. Our answers:

Reviewer 1

1.The last sentence of the introduction (lines 89-91) may be omitted.  It adds nothing new and raises doubts about the results obtained in the work. 

The last sentence is altered as follows:

“Our findings reported in this paper indicate that the involvement of Mer in adipose tissue homeostasis is more complex than we expected, because Mer is also expressed by adipocytes, and loss of adipocyte Mer dominates the resulting phenotype.”

  1. 2. In results, lines 266-267. Since the values of weight changes have been described in body weight, the numerical value should also be express in the change in weight of gonadal fat tissue.

The requested data are now included into the text.

  1. Figure 2a shows that HFD significantly inhibits Mer expression in Mer +/+ gWAT macrophages and the figure caption is Mer loss protects mice against HFD-induced inflammation, hepatosteatosis and insulin resistance. We know that HFD induce inflammation in adipose tissue. How does authors explain this data?

What Figure 2 shows is that loss of Mer in the full body protects mice against obesity (mainly via loss of Mer from the adipose tissue), that is why we do not see signs of inflammation in the Mer null adipose tissue. The reason is (as we show) that in the absence of Mer the thermogenesis is activated in the brown adipose tissue releasing the excess calorie taken up in the form of heat. The lower expression of Mer in adipose tissue macrophages on high fat diet is not necessarily the result of a downregulation. It rather reflects the change in the macrophage populations. While in non-obese mice the macrophages are tissue resident macrophages known to express high levels of Mer, on HFD pro-inflammatory macrophages infiltrate the adipose tissue in high amounts, which express low levels of Mer (Zagorska et al. Nat. Immunol. 2014. 15:920). To make this more clear we altered the text as follows:

“HFD significantly enhanced the mRNA expressions of several pro-inflammatory cytokines, as well as that of resistin in the Mer+/+ ATMs as well, while the Mer mRNA expressions were significantly lower. In contrast, significantly less or no alterations were seen in Mer-/- MΦs exposed to HFD (Figure 2A). mRNA expression alterations in the wild-type ATMs reflect the change in the composition of MFs in obesity: while in non-obese mice the ATMs are tissue resident MFs known to express high levels of Mer but low levels of pro-inflammytory cytokines, on HFD pro-inflammatory MFs infiltrate the adipose tissue, which express low levels of Mer [26].”

  1. Figures 2b and 2c show liver weights and liver TAG content. In Mer +/+ mice HFD significantly increases TAGs but liver weights do not change. How does authors explain this data?

We were also surprised, because our similar studies on HFD previously indicated also a liver weight increase (Sághy et al. Cell Death Dis. 2019). My only explanation is that we keep these mice in heterozygous form, but not in a terribly big group. If there was a mutation that is generated and by now seen in all mice, and which affects the viability of hepatocytes while processing high amount of carbohydrate and lipids, could cause this result. We did not investigate the reason.

  1. In figure 3 should order the letters of the graphics

The reason, why we selected this order originally was to place the data related to each other closer. However, we changed the other now, as you suggested.

  1. In section 3.4 authors evaluate the expression of TAM kinases. Authors should explain better the importance of these kinases

At this point we only wanted to demonstrate that Mer is expressed by adipocytes, but not only Mer, also the other two members of the TAM kinase family. The role of Tyro3 was not investigated yet. The Axl paper came out during the time we performed these experiments. We are referring to the phenotype of the adipocyte specific Axl knock out mice in the paper.

Reviewer 2 Report

Comments and Suggestions for Authors

In the paper: Loss of MER tyrosine kinase attenuates adipocyte hypertrophy and leads to enhanced thermogenesis in mice exposed to high-fat diet, Köröskényi et al. investigated the role of Mer tyrosine kinase in DI obesity. They found that the loss of Mer protects mice from HFD-induced obesity. While Mer activity in mice exposed to HFD promotes lipid storage in adipocytes both in vitro and in vivo, it simultaneously attenuates thermogenesis in BAT, contributing to its 'whitening', at least at the histological level. Conversely, attenuating this inhibition in Mer null mice prevented HFD-induced whitening and increased gene expression of several targets related to lipid metabolism and thermogenesis in BAT but not in gWAT. The authors therefore concluded that abrogating Mer-mediated inhibition in HFD may be a promising approach in DIO to prevent lipotoxicity and hepatic steatosis. I find the results interesting and relevant for obesity research and therefore recommend them for publication in Cell's. Nevertheless, I have a few comments that should be resolved:

My comments:

1. Introduction: one gets the impression that the paper examines the role of efferocytosis and EVs in regulating fat metabolism (which is emphasised in the introduction, especially in the last few sentences), but in fact none of this was examined directly, only markers of fat metabolism, depending on Mer expression throughout the body. This is too mechanistic and realistically there is no clear evidence that this metabolic pathway is involved. Therefore, in my opinion, the hypothesis should be weakened/reworded.

2. The same applies to the discussion in which the expression of Mer is unfoundedly linked to the regulation of the Akt/MTORC1 signalling pathway, where no molecular target of this pathway has been investigated and demonstrated. It is necessary to either add results for this signalling pathway or to weaken the numerous conclusions in the discussion that directly refer to the AKT signalling pathway.

3. In this context, the part of the results referring to the mTORC1 signalling pathway (lines 401-407) should be transferred to the Discussion and this part of the Discussion should be reduced to a reasonable level, as this gives the impression that it has been both investigated and shown in the results.

4. The extent to which the pro-inflammatory state is related to browning in the WAT is clear from the literature, which would need to be discussed in the context of the Mer expression.

5. The sentence (lines 575-577) should be more precise (precise Tyr kinase signalling) and supported by a reference (see "previous studies")?

6.Material and methods: Line 173. was beta-actin used for normalisation? Please check.

7. Material and methods: Line 210. text duplication

8. Material and Methods: Line 252. what does **** mean? Add text.

9.Results: Please check the magnification of the microscopic images (Fig. 1), as it appears that the adipocytes in the HFD groups are very large and have proportionally large cytoplasmic areas, suggesting greater magnification.

10. Was plasma insulin (as in the M&M section, line 184) or serum insulin (as in the Results and Discussion, lines 334, 342) analysed?

11. Uniform reference style, line 423

12. In Figure 7, two-way ANOVA should be used for cAMP and UCP1 analysis.

13. Abbreviations should be used if they are mentioned for the first time in the text, otherwise the full names (PGC, PPAR, IGF-1, etc.).

14. Check the spelling for editorial errors (antibidy, line 521...).

Author Response

We would like to thank the reviewers for their careful work with which they strongly increased the clarity and the value of our paper, and we hope that they can accept our paper in the revised version. Our answers:

Reviewer 2

1.Introduction: one gets the impression that the paper examines the role of efferocytosis and EVs in regulating fat metabolism (which is emphasised in the introduction, especially in the last few sentences), but in fact none of this was examined directly, only markers of fat metabolism, depending on Mer expression throughout the body. This is too mechanistic and realistically there is no clear evidence that this metabolic pathway is involved. Therefore, in my opinion, the hypothesis should be weakened/reworded.

The original goal was to investigate these processes, but naturally we did not continue to do it, once we saw that the phenotype, we found, does not depend on the hematopoetic cells (see the results of the bone marrow transplantation experiments). To make this point more clear, we altered the text in the results section as follows:

“Thus, instead of investigating further the role of Mer in MFs as we planned originally, we decided to determine whether TAM kinases are expressed by adipocytes of the gWAT in mice”.

There is only one result that indicates we might be right with our original hypothesis, the higher adipocyte size on ND in those bone marrow transplanted mice in which Mer is missing in the bone marrow derived cells. In the Discussion we talked about this point:

In the presented experiments, the involvement of Mer tyrosine kinase was investigated in the maintenance of adipose tissue homeostasis. Since MFs are known to contribute to adipose tissue homeostasis, and Mer is highly expressed by tissue-resident MFs, our prediction was that Mer acting in the myeloid compartment would have a protective role against HFD-induced adiposity and inflammation. Indeed, bone marrow transplantation experiments in which Mer was missing from the hematopoietic cells indicated enhanced adipocyte hypertrophy in mice kept on both ND and HFD. The limitation of these studies was the strong negative influence of irradiation on the ability of mice to gain body weight. Thus to address the precise role of Mer in the adipose tissue homeostasis played in the myeloid compartment, mice carrying myeloid-specific deletion of Mer should be investigated in the future.”

  1. The same applies to the discussion in which the expression of Mer is unfoundedly linked to the regulation of the Akt/MTORC1 signalling pathway, where no molecular target of this pathway has been investigated and demonstrated. It is necessary to either add results for this signalling pathway or to weaken the numerous conclusions in the discussion that directly refer to the AKT signalling pathway.
  2. In this context, the part of the results referring to the mTORC1 signalling pathway (lines 401-407) should be transferred to the Discussion and this part of the Discussion should be reduced to a reasonable level, as this gives the impression that it has been both investigated and shown in the results.
  3. The extent to which the pro-inflammatory state is related to browning is clear from the literature, which would need to be discussed in the context of the Mer expression.

We accept these points and altered the text accordingly as follows:

“Unexpectedly, Mer is also expressed by adipocytes, and the simultaneous loss of Mer in adipocytes of the full-body knock-out mice overwrote the myeloid-specific Mer null phenotype. In the absence of Mer not only the hypertrophy of white adipocytes is attenuated but Mer has such a strong inhibitory effect on the thermogenesis in the BAT that loss of it led to a sufficient increase in the rate of thermogenesis to prevent lipid accumulation in the Mer null mice kept on HFD diet. Our data indicate that the observed enhanced thermogenesis of brown adipocytes in the absence of Mer might be related to the regulation of intracellular cAMP levels by Mer, similar to the other TAM kinase Axl, which was reported to inhibit cAMP phosphodiesterases in brown adipocytes via activation of AKT [51]. Similar to Axl, Mer is also known to activate AKT [52].

The lack of white adipocyte hypertrophy in Mer null mice exposed to HFD could be the result solely due to the enhanced thermogenesis detected in the BAT. Our results conducted on the 3T3-L1 adipocyte cell line, however, indicate that Mer can directly affect the hypertrophy of adipocytes exposed to fatty acids. The signaling pathway via which Mer promotes the hypertrophy of adipocytes was not investigated in our study. However, the mechanistic target of rapamycin (mTOR), and especially mTORC1 has been implicated to play a crucial role in promoting adipocyte growth and fat accumulation by two other tyrosine kinases, insulin and insulin-like growth factor receptor 1 (IGFR1) [3, 53]. Our data obtained here also indicate that the Mer-induced adipocyte hypertrophy pathway is not WAT specific, as it promotes lipid accumulation also in BAT adipocytes of wild-type mice exposed to HFD leading to whitening of the BAT tissue [49,50]. Fat accumulation in the BAT similar to WAT triggers pro-inflammatory macrophage recruitment, and the pro-inflammatory cytokines released might also contribute to the observed inhibition of thermogenesis [54]. “

  1. The sentence (lines 575-577) should be more precise (precise Tyr kinase signalling) and supported by a reference (see "previous studies")?

These previous studies were listed in the following sentences.

“The fact that the intensity of the tyrosine kinase signaling pathways in adipocytes plays a decision role between storing excess lipids or converting it to heat, has already been indicated by previous studies. Thus, the injection of a single dose of insulin decreased the expression of PGC1a and UCP1 in BAT [57]. Accordingly, mice generated by deletion of one of the insulin alleles resulting in lower diet-induced insulin levels were resistant to diet-induced obesity and showed increased browning of WAT [58]. Interestingly, not only the tyrosine kinase pathway triggered by insulin might contribute to this metabolic decision, since mice characterized by adipocyte-specific deletion of insulin-like growth factor receptor 1 [59], as well as that of Axl [54] showed also protection against HFD-induced obesity and enhanced BAT thermogenesis in mice.

6.Material and methods: Line 173. was beta-actin used for normalisation? Please check.

Thank you for noticing. It is beta-actin. When I transferred the text into the MDPI format the symbol label was lost.

  1. Material and methods: Line 210. text duplication

Thank you for noticing. Text duplication is removed now.

  1. Material and Methods: Line 252. what does **** mean? Add text.

Text is altered as follows:

“For datasets split on two independent factors, two-way ANOVA (multiple comparison procedures with Tukey post hoc test) was used.* denotes p 0.05, ** denotes p 0.01, *** denotes p 0.001, **** denotes p0.0001.”

9.Results: Please check the magnification of the microscopic images (Fig. 1), as it appears that the adipocytes in the HFD groups are very large and have proportionally large cytoplasmic areas, suggesting greater magnification.

The magnification was the same in each case. The scale bar is automatically added to the picture.

  1. Was plasma insulin (as in the M&M section, line 184) or serum insulin (as in the Results and Discussion, lines 334, 342) analysed?

Thank you for noticing! It was serum insulin as correctly stated in the Results section.

  1. Uniform reference style, line 423

Thank you for noticing! Corrected.

  1. In Figure 7, two-way ANOVA should be used for cAMP and UCP1 analysis.

Thank you for noticing. Two way ANOVA was used.

  1. Abbreviations should be used if they are mentioned for the first time in the text, otherwise the full names (PGC, PPAR, IGF-1, etc.).

Checked and changed accordingly.

  1. Check the spelling for editorial errors (antibidy, line 521...).

Checked and corrected.

Reviewer 3 Report

Comments and Suggestions for Authors

The research focuses on the role of MER tyrosine kinase on adipose tissue hypertrophy and thermogenesis. The study is interesting, but the introduction, results, figures, and discussion must be revised before publishing.

  1. Introduction: The lipid accumulation is associated with enhancing leptin and resistin but decreasing adiponectin, which are factors for insulin resistance and diabetes. However, this study does not find a link between lipid accumulation and MER tyrosine kinase. Why do the authors choose MER tyrosine kinase among many receptors but not other receptors such as LOX-1 (lectin-like oxidized LDL receptor 1), scavenger receptor class B type 1 (SRB1), and other toll-like receptors? Can the authors provide omic data showing MER tyrosine kinase is the most important and associated with adipocyte hypertrophy?
  2. Introduction or Results: Can a high-fat diet impair MER gene and MER protein expression? Do the authors have data to support this hypothesis? What is the ligand of MER? What is the upstream mechanism?
  3. Introduction: It is inappropriate for the conclusion to be presented in the introduction section. "Our findings reported in this paper indicate that the involvement of Mer in adipose tissue homeostasis is more complex than we expected."
  4. Introduction: I don't think there is a strong rationale to hypothesize: "Since the lipid-containing extracellular vesicles released by adipocytes are also PS positive [8], Mer might also contribute to the lipid vesicle uptake by adipose tissue MΦs [32]."
  5. Materials and Methods/ Figure 2: The authors must revise the study design of "Bone marrow transplantation." Need to rewrite the figure legend. Need to define -/- --> +/+. Any western blot or IHC of MER expression in the livers of HFD mice? How do the authors define controls (positive or negative control for transplantation), and how do the authors consider successful transplantation?
  6. Figure 1, Figure 2, and all: The authors must present genes in Italic, ie. Figure 1G, IL-6; TNF-α, MCP1...etc. MCP-1 or MCP1? Make it consistent. The authors need to clarify the used statistics. The authors must use ## when comparing HFD Mer+/+ and HFD Mer-/-. Mer+/+ and Mer-/- must be in Italic.
  7. Figure 2: The authors must quantify the lipid droplet area and the NAFLD score. Figure 2D shows a dramatic difference. However, n=3 is not enough. We need to see all the raw data; the liver tissue figures must be more than one for each sample. Can the authors provide raw data in three high-power fields for each sample? Can calculate the lipid droplet area and the NAFLD score.
  8. Figures 1, 2, and 4: the Y-axis is wired. Target rel. Gene expre. The authors need to follow the general rules of presentation. 
  9. Figure 4: Why Mer-/- decreases Tyro3/Axl? Any agonist or positive/negative inhibitor controls?
  10. Figure 5: the comparison must always focus on HFD Mer+/+ vs. HFD Mer-/-. How do the authors interpret Figure 5C? Is there any evidence to support that Mer impairs mitochondrial function?
Comments on the Quality of English Language

Quality of English Language is ok.

Author Response

We would like to thank the reviewers for their careful work with which they strongly increased the clarity and the value of our paper, and we hope that they can accept our paper in the revised version. Our answers:

Reviewer 3

1.Introduction: The lipid accumulation is associated with enhancing leptin and resistin but decreasing adiponectin, which are factors for insulin resistance and diabetes. However, this study does not find a link between lipid accumulation and MER tyrosine kinase. Why do the authors choose MER tyrosine kinase among many receptors but not other receptors such as LOX-1 (lectin-like oxidized LDL receptor 1), scavenger receptor class B type 1 (SRB1), and other toll-like receptors? Can the authors provide omic data showing MER tyrosine kinase is the most important and associated with adipocyte hypertrophy?

We did not select Mer based on omic data. We were interested originally, how the loss of a determinant macrophage efferocytosis receptor known to have also strong anti-inflammatory and potential lipid EV uptake activities would affect the adipose tissue homeostasis. The findings, we report, were unexpected, since we found Mer to be expressed by adipocytes as well. We believe that not the expression of Mer but that of its ligand is related to human obesity, as we mention it in the discussion.

“obesity in humans is associated with increased plasma Gas6 levels”

2.Introduction or Results: Can a high-fat diet impair MER gene and MER protein expression? Do the authors have data to support this hypothesis? What is the ligand of MER? What is the upstream mechanism?

We believe that the decreased Mer tyrosine kinase expression observed in adipose tissue macrophages on high fat diet is not the result of a downregulation. It rather reflects the change in the macrophage populations. While in non-obese mice the macrophages are tissue resident macrophages known to express high levels of Mer, on HFD pro-inflammatory macrophages infiltrate the adipose tissue in high numbers, which express low levels of Mer (Zagorska et al. Nat. Immunol. 2014. 15:920). We altered the text to make these data more clear as follows:

HFD significantly enhanced the mRNA expressions of several pro-inflammatory cytokines, as well as that of resistin in the Mer+/+ ATMs as well, while the Mer mRNA expressions were significantly lower. In contrast, significantly less or no alterations were seen in Mer-/- MΦs exposed to HFD (Figure 2A). mRNA expression alterations in the wild-type ATMs reflect the change in the composition of MFs in obesity: while in non-obese mice the ATMs are tissue resident MFs known to express high levels of Mer but low levels of pro-inflammytory cytokines, on HFD pro-inflammatory MFs infiltrate the adipose tissue, which express low levels of Mer [26].

The known ligands of Mer tyrosine kinase are protein S and Gas6, as it is stated in the introduction part.

3.Introduction: It is inappropriate for the conclusion to be presented in the introduction section. "Our findings reported in this paper indicate that the involvement of Mer in adipose tissue homeostasis is more complex than we expected."

This is incorrect. In fact, many journals request to give a short summary of the results at the end of the introduction. One example from the prestigious journal Cell Death and Differentiation.

Wu et al. Ubiquitin ligase E3 HUWE1/MULE targets transferrin receptor for degradation and suppresses ferroptosis in acute liver injury. CDD 29: 1705–1718 (2022)

4.Introduction: I don't think there is a strong rationale to hypothesize: "Since the lipid-containing extracellular vesicles released by adipocytes are also PS positive [8], Mer might also contribute to the lipid vesicle uptake by adipose tissue MΦs [32]."

Then read, please, in: A review of the regulatory mechanisms of extracellular vesicles-mediated intercellular communication. Cell Commun Signal 2023 Apr 13;21(1):77.

 “Similarly, phosphatidylserine is indirectly recognized by the growth arrest-specific protein 6, Gas6. The phosphatidylserine-Gas6 complex activates the MER tyrosine kinases on the surface of macrophages which triggers the EV uptake and causes an anti-inflammatory phenotype.”

5.Materials and Methods/ Figure : The authors must revise the study design of "Bone marrow transplantation." Need to rewrite the figure legend. Need to define -/- --> +/+. How do the authors define controls (positive or negative control for transplantation), and how do the authors consider successful transplantation?

We setup originally the transplantation protocol using C57Bl/6 and BoyJ mice. This experimental BMT CD45 congenic model allowed us to detect donor, competitor and host contributions in hematopoiesis and repopulation efficiency of donor cells (congenic mice with CD45.1 versus CD45.2). The CD45.1 and CD45.2 contribution were then detected by flow cytometry 8–12 weeks following BMT usually with more than 95% repopulation efficiency (Sághy et al. CDD 2019,10:439). However, the irradiation very strongly prevents weight gain, and BoyJ mice are not so good in gaining weight anyway, that is why we used only the Mer+/+ and Mer-/-  mice for these experiments with the same BMT protocol, we know works in our hands. These mice are exposed to such a high level of full body irradiation that without successful transplantation they die. So the success means that they survive and gain weight. In these experiments the controls were those mice in which wild-type bone marrow was transplanted into wild-type irradiated mice (we added this sentence to the text).

Explanation is added now to the figure legend 3, how we defined the mice.

Arrows indicate, which type of bone marrow cells are transplanted into which genetic background type of host animals.”

6.Figure 1, Figure 2, and all: The authors must present genes in Italic, ie. Figure 1G, IL-6; TNF-α, MCP1...etc.. The authors must use ## when comparing HFD Mer+/+ and HFD Mer-/-. Mer+/+ and Mer-/- must be in Italic.

This is not used this way generally. Just one example from https://doi.org/10.1038/icb.2014.97

“Mild EAO was observed in Axl−/− or Mer−/− mice. Notably, severe EAO at stages 4 and 5 were developed in immunized Axl−/−Mer−/− mice. According to EAO severity, testis weight significantly decreased in immunized Axl−/−Mer−/− mice (Figure 1c). Immunization failed to significantly alter testis weight in WT, Axl−/− and Mer−/− mice.”

The authors need to clarify the used statistics.

The name of the statistics identifies the type of statistics used and explains why. For every pair compared (ANOVA provides comparison for every pair) only the significant differences are shown with the code given in the Materials and Methods section (* denotes p 0.05, ** denotes p 0.01, *** denotes p 0.001, **** denotes p0.0001).

MCP-1 or MCP1? Make it consistent

MCP1 is converted now to MCP-1 in Figure 2A. Thank you for noticing.

7.Figure 2: The authors must quantify the lipid droplet area and the NAFLD score. Figure 2D shows a dramatic difference. However, n=3 is not enough. We need to see all the raw data; the liver tissue figures must be more than one for each sample. Can the authors provide raw data in three high-power fields for each sample? Can calculate the lipid droplet area and the NAFLD score.

Any western blot or IHC of MER expression in the livers of HFD mice?

Mer is known to be expressed by Kupffer cells and capillary endothelial cells in the liver.

“Differential regulation of hepatic physiology and injury by the TAM receptors Axl and Mer.”

DOI 10.26508/lsa.202000694

|

We were not interested truly in the liver in details. We just wanted to demonstrate that in wild-type livers we see lipid accumulation, as expected (not novel), while in the Mer null mice not, when fed on HFD. The H & E pictures are so striking that they speak for themselves. The quantification was based on determining the triacylglycerol content of the livers, and for this analysis we used livers from 5 mice, as indicated in the Figure legends. But for the interest of the reviewer, we add here the requested pictures.

8.Figures 1, 2, and 4: the Y-axis is wired. Target rel. Gene expre. The authors need to follow the general rules of presentation. 

We do not understand what the problem is with these figures. These data show relative expression of the mRNA of the indicated gene compared to the beta-actin mRNA, as it is described in the Materials and Methods. We regularly publish mRNA expression data in this form.

9.Figure 4: Why Mer-/- decreases Tyro3/Axl? Any agonist or positive/negative inhibitor controls?

Loss of Mer did not affect the basal levels of Tyro3 or Axl in the adipocytes of gWAT, as shown in Figure 4.

10.Figure 5: the comparison must always focus on HFD Mer+/+ vs. HFD Mer-/-.

In the two way ANOVA we compare everybody to everybody but show only those comparisons, which are significantly different. And all the differences can be interesting.

How do the authors interpret Figure 5C?

To answer your question, we added to the text:

“We found that the expression of lipases was significantly decreased in the HFD-exposed Mer+/+ gWAT, while no such alterations were detected in the Mer-/- gWAT indicating that while in Mer+/+ adipocytes the triacylglycerol degradation might have been attenuated following exposure to HFD contributing to the adipocyte hypertrophy, it was not altered in the Mer-/- adipocytes”.

Is there any evidence to support that Mer impairs mitochondrial function?

So far I have not seen data for it.

PS: I copied the liver histology section into the text. But they did not go through with the text when I copied them into this frame. I triy to appload a pdf version as well.
